# The Short- and Long-Term Impact of COVID-19 Lockdown on Child Maltreatment

**DOI:** 10.3390/ijerph19063350

**Published:** 2022-03-12

**Authors:** Mengqing Long, Jia Huang, Yishun Peng, Yawen Mai, Xian Yuan, Xinhua Yang

**Affiliations:** 1Department of Psychology, Hunan Agricultural University, Changsha 410128, China; longmengqing2009@hunau.edu.cn; 2Neuropsychology and Applied Cognitive Neuroscience Laboratory, CAS Key Laboratory of Mental Health, Institute of Psychology, Chinese Academy of Sciences, Beijing 100101, China; huangj@psych.ac.cn; 3Department of Psychology, The University of Chinese Academy of Sciences, Beijing 100101, China; 4Shanghai Changning Mental Health Center, Shanghai 200335, China; pys19961230@163.com (Y.P.); miya06011995@163.com (Y.M.); yuanxian2022@163.com (X.Y.); 5Centre for Affective Disorders, Psychological Medicine, Institute of Psychiatry, Psychology & Neuroscience, King’s College London, London SE5 8AF, UK

**Keywords:** child maltreatment, sexual abuse, lockdown, social anhedonia, psychotic symptoms

## Abstract

Background: The COVID-19 pandemic has brought a new threat to child health and safety. Some studies suggest that social isolation and economic stress have exacerbated child abuse and neglect, whereas other studies argue that orders to stay at home are likely to promote parent–child relationships during this stressful time. Due to a lack of prospective studies including before–during–after lockdown assessments, the impacts of lockdown measures on child maltreatment are unclear. Methods: This study retrospectively investigated child maltreatment of 2821 Chinese children and adolescents from 12 to 18 (female, 59%) before, during and after lockdown, and identified risk factors. Potential predictors including socio-economic and individual mental health status were collected. Results: During Chinese lockdown, children and adolescents reported that the proportions of decrease (range 18–47.5%) in emotional abuse and neglect, physical abuse and neglect, sexual abuse, and witnessing domestic violence were greater than that of increase (range 5.1–9.1%). Compared with before lockdown (1.6%), the prevalence of sexual abuse significantly increased 8 months (2.9%) after the lifting of lockdown (*p* = 0.002). Being male, suffering from depression, state anhedonia, and experiencing psychotic symptoms at baseline were associated with increased sexual abuse after lockdown. Conclusions: The impact of lockdown on child maltreatment was beneficial in the short-term but detrimental in the long-term in China.

## 1. Introduction

Children are the most vulnerable population during disasters and emergencies. The COVID-19 pandemic, as a public health crisis, has brought a new threat to child health and safety. National lockdowns may fuel parental tension and then lead to an increase in child maltreatment, including child abuse and neglect, and domestic violence. Historically, all types of child abuse have soared following natural disasters and economic downturns, while limited access is available to the usual services that provide support [1]. News reports of increased rates of hospital visits and hospitalizations for child abuse attributed to the COVID-19 pandemic have already surfaced [2]. In this situation, scholars worldwide are worrying that children may be exposed to a significant risk for maltreatment [3,4]. Indeed, research on COVID-19 was beginning to show lockdown restrictions increased the incidence of child abuse and neglect relative to pre-lockdown levels [5,6,7]. Parents experiencing job losses, social isolation and negative parenting behaviours were at high risk to engage in psychological and physical abuse [8,9,10]. However, these cross-section studies examined the short-term impact of lockdown measures, and did not provide a longer period of observation after lockdown.

Despite the substantial effects of lockdowns on everyday life, one potential positive outcome of this pandemic was the promotion of wellbeing for parent–child relationships during this stressful time [11]. For example, the majority of parents engaged in more everyday activities with their child [12], showed more physical affection, warmth, and love toward their children [13], and experienced better relationships with family members during lockdown [14,15]. There was prior evidence that disasters and pandemics can stimulate social connectedness and closeness [16]. Specifically, after the severe acute respiratory syndrome epidemic in 2003, people in Hong Kong reported increased feelings of embeddedness in the community and caring for friends and family members [17]. People valued their family and re-evaluated the importance of connectedness after the Great East Japan Earthquake [18]. Indeed, a recent systematic review of longitudinal studies revealed that the impact of COVID-19 pandemic lockdowns on mental health was small in magnitude but had no effect on positive functioning, such as wellbeing, life satisfaction, and social support, among the general population [19]. Similarly, school closures were not detrimental, but were beneficial for children and adolescents, showing a reduction in anxiety and an increase in connectedness [20]. Most students were satisfied with their life after school closures, and 21.4% even reported higher life satisfaction than before the pandemic [21].

Taken together, some scholars argued that lockdowns may exacerbate children’s vulnerability to maltreatment at home due to increased rates of poverty, food insecurity, unemployment, and inequalities [3,4]. Although there has been abundant speculation about the impact of COVID-19 lockdowns on child maltreatment, few longitudinal data support these assumptions. On the other hand, a resilience perspective suggested that the pandemic may make positive changes, such as family cohesion and closeness [11]. Consequently, investigating how child maltreatment changed throughout the COVID-19 lockdown (from before and during, to after lockdown) will provide valuable insights into the public health system; in particular, whether it worsened or improved during lockdown, whether it was affected by lockdown measures, whether it was unchanged or returned to normal after lockdown restrictions were eased, and which factors predicted these short-term and longer-term changes.

This prospective study aimed to investigate how child maltreatment was affected by lockdown among children and adolescents. China provided a unique context to investigate these questions because of its success in eliminating community transmission, which allowed people to return to pre-COVID life. Based on the association between increased child maltreatment and natural disasters, the current study hypothesized that child maltreatment increased during the lockdown but recovered to pre-lockdown levels after the lifting of the lockdown. The secondary aim was to identify potential predictors for child maltreatment during the COVID-19 crisis, including exploring: (i) which socio-demographic characteristics, such as age, gender, place of residence, household income, parental employment, and marital relationship, were increased risk factors for child maltreatment; and (ii) whether mental health symptoms, such as anhedonia, depression, anxiety, and psychotic symptoms, can explain increases in child maltreatment. Given child maltreatment was previously linked to negative mental health symptoms, economic pressure, and family characteristics [22], we further hypothesized that these symptoms and demographic variables would predict the increase in child maltreatment.

## 2. Materials and Methods

### 2.1. Participants

The two-wave study was conducted among a convenient sample from three secondary schools at Changsha city in Hunan province. At baseline, students from all participating schools were invited to participate in the study. Of the 3273 invited students, a total of 2821 (86.19%) children and adolescents participated in the first survey from 22 September to 25 October 2020, 5 months after the lifting of Chinese lockdown (started on 23 January 2020, ended on 8 April 2020). Participants completed the child maltreatment history questionnaire related to the lockdown. Eligible participants in the second survey were those who had participated at baseline. The second survey (n = 2470) was conducted 8 months after lockdown, from 29 December 2020 to 16 January 2021. The prospective mental health assessments, followed by a retrospective maltreatment report, have significant methodological advantages, including assessing the mental health consequences of maltreatment in an adolescent sample [23]. The following exclusion criteria were applied: (1) students or their guardian (n = 452) were not willing to participate in the study; (2) 122 students were inclined to respond to the items in a similar pattern (e.g., chose the same answer within the whole questionnaire); (3) 94 participants answered twice and their second answers were excluded; (4) 141 participants were excluded because the completion time was shorter than 554 s (2.5th percentile) or longer than 4204 s (97.5th percentile). Valid participants in the final sample (n = 2464, 75.28%) were those who completed each questionnaire at each timepoint, comprising 1461 (59%) females and 1003 (41%) males aged from 12 to 18, with a mean age of 15.48 (SD = 1.76).

#### 2.1.1. Sociodemographic Variables

The following variables were chosen based on their association with childhood maltreatment: (1) gender, (2) age, (3) place of residence (urban, rural), (4) monthly household income (low, medium, high); (5) father employment (job, jobless); (6) mother employment (job, jobless); (7) parental marital relationship (normal, separation, divorce); (8) COVID-19 diagnosis (yes diagnosed, suspected, don’t know, no).

#### 2.1.2. Child Maltreatment

The Childhood Trauma Questionnaire-Short Form (CTQ-SF) was used to assess child maltreatment history at home [24]. It was a 28-item retrospective self-report questionnaire with five subscales: emotional neglect, physical neglect, sexual abuse, emotional abuse, and physical abuse. Each item was measured on a 5-point Likert scale ranging from 1 (not at all) to 5 (very often). The cut-offs of each subscale were emotional abuse ≥13, emotional neglect ≥15, sexual abuse ≥8, physical abuse ≥10, and physical neglect ≥10. Six additional items were added as a measure of witnessing domestic violence and severe physical abuse from parents during lockdown: were your parents physically abusive?; were your parents emotionally abusive?; were your parents violent toward his/her siblings?; were you burned by your parents?; were you threatened by your parents using a knife or gun?; were you choked by your parents?. The CTQ response scale typically asks about the occurrence of behaviours within the past year. In this study, participants were asked to retrospectively assess child maltreatment experiences that occurred before, during, and after lockdown. At baseline, they were asked to report child maltreatment experiences in their childhood prior to lockdown and changes in child maltreatment experiences during lockdown, for which the response scale was modified to ask how changes in behaviours occurred “during lockdown” (1 = decreased, 2 = unchanged/same as before, 3 = increased). At follow-up, participants were again asked to report child maltreatment experiences from the lifting of lockdown to the present (over an 8-month period). The 34-item scale was found to have good internal consistency (Cronbach’s α = 0.72) in the current study.

#### 2.1.3. Depressive Symptoms

The short version of the Mood and Feelings Questionnaire (MFQ-13) was used to measure depressive symptom severity during the past two weeks [25]. The inventory consisted of 13 items, each of which were answered on a 3-point Likert scale (score range: 0–26), with higher scores representing a higher occurrence of symptoms associated with depression. A cut-off score of 8 is usually used for demonstrating clinically depressive symptoms.

#### 2.1.4. Anxiety Symptoms

The Generalized Anxiety Disorder 7-item scale (GAD-7) was used to assess the symptoms of anxiety during the previous two weeks [26]. Each item ranged from 0 (not at all) to 3 (nearly every day). A cut-off score of 9 is usually used for demonstrating clinically anxious symptoms.

#### 2.1.5. Psychotic Symptoms

The Brief 8-Item Community Assessment of Psychotic Symptoms-Positive Scale (CAPE-P8) was used to assess subclinical positive psychotic symptoms [27]. Among these items, six items were related to delusions and two were related to hallucinations. Responses to items ranged from 1 (never) to 4 (nearly always), with higher scores reflecting higher levels of psychotic symptoms.

#### 2.1.6. Anhedonia

The Chinese adolescent version of the Anticipatory and Consummatory Interpersonal Pleasure Scale (ACIPS-A) was used to assess social pleasure experiences [28]. The measure was rated on a 4-point Likert scale from 0 (totally false for me) to 4 (totally true for me), with lower scores indicating a greater likelihood of social anhedonia.

The Snaith–Hamilton Pleasure Scale (SHAPS) was used to assess general pleasure experiences in the last few days [29]. The SHAPS consisted of 14 items each answered on a 4-point Likert scale, with higher scores reflecting higher levels of state anhedonia.

### 2.2. Procedure

Because the Chinese Government still recommended that the public minimized face-to-face interaction after lockdown, this study was completed through an online survey platform (www.wjx.com, baseline: from 22 September to 25 October 2020; follow-up: from 29 December 2020 to 16 January 2021), with the relevant smartphone link being pushed to students and their guardians via a family–school communication app by school mental health services. Clicking on the link to the survey guided potential respondents to a page that provided information about the purpose of the study, the nature of the questions, and two electronically informed consent forms for students and their guardians, respectively. To reduce the potential possibility of parental intervention in reporting child maltreatment, the survey was conducted during a regularly scheduled class period for mental health activity at school. Before the start of the investigation, the details of the study were given again by a psychological teacher to all responders. Participation was voluntary and no incentive reward was given. Anonymity was emphasized, but as the survey was a longitudinal study, participants were asked to give the last four digits of their Student Identification number. It took participants about 30 min to complete the survey. Ethical approval was obtained from the Institutional Review Board of the Wenzhou Medical University (2020-131).

Child maltreatment history was retrospectively assessed pre-, during and post-lockdown. At baseline, participants were asked to report their child maltreatment experience before lockdown via the Childhood Trauma Questionnaire. To assess changes in child maltreatment related to the lockdown, participants also were asked to report their maltreatment experience for each item in terms of unchanged, increased, and decreased during lockdown as compared to before. At follow-up, participants were asked to report their child maltreatment experience, using the Childhood Trauma Questionnaire, from the lifting of lockdown to the present. Participants at baseline were also asked to retrospectively report depression, anxiety, and psychotic symptoms during lockdown.

### 2.3. Data Analysis

First, Little’s (1988) missing completely at random (MCAR) test was initially conducted to understand the nature of our missing data with regard to total scores over time [30]. Independent sample t-tests were conducted to examine whether adolescents who completed two waves of data collection differed from those with missing data. Then, descriptive analysis presented the proportions of individuals reporting a decease, an increase, or unchanged at the item level of the mild maltreatment measure, and cross-sectional univariate regression analysis was performed to determine which factors, including socio-demographic variables and depression, anxiety, and psychiatric symptoms of the lockdown period, were associated with changes in child maltreatment during lockdown. Chi-square tests with Bonferroni correction were further conducted to compare the prevalence of 5 types of maltreatment experiences pre- and post-lockdown. Finally, the linear mixed model was used to examine the potential impact of lockdown on change within an individual in child maltreatment experiences. Interactions between the time (pre- to post-lockdown) and the six socio-demographic subgroups were used to investigate differences in changes associated with the pandemic between different populations. Interactions between the time and mental health status were used to investigate which psychological factors were risk factors for changes in child maltreatment. These variables were gender, place of residence, household income, parental employment status, parental relationship, depression, anxiety, psychotic symptoms, state anhedonia, and social anhedonia. A positive coefficient from the mixed-effects model indicates worsening child maltreatment associated with the pandemic. Age was controlled for in the model because maltreatment changed with age throughout adolescence. To avoid problems associated with multicollinearity, the continuous variables were mean centered. Bonferroni correction was used to correct multiple comparison. Statistical analysis was performed using SPSS 27.0. 

## 3. Results

Little’s MCAR test suggested that listwise deletion of participants with incomplete follow-up assessment data was appropriate, χ^2^(28) = 31.72, *p* = 0.29. Groups did not differ in depression [t(2819) =1.53, *p* = 0.13], anxiety [t(2819) = −0.30, *p* = 0.77], psychotic symptoms [t(2819) = 0.71, *p* = 0.48], and child maltreatment [t(2819) = 1.31, *p* = 0.19] at baseline between the final sample (n = 2464) and those who did not complete the following assessment. Those with missing values (357, 14%) for variables were therefore excluded.

### 3.1. Child Maltreatment during Lockdown

Table 1 presents the proportions of participants at the item level for child maltreatment during lockdown. From this it can be seen that the proportions of decreases in child maltreatment were greater than those of increases. Overall, a range from 18% to 47.5% of participants reported that they experienced decreased abuse and neglect experiences during lockdown. Only a small proportion (range 5.1–9.1%) of participants reported an increase in child maltreatment experiences. The majority of participants (range 46.3–72.7%) did not feel any changes due to the lockdown. Additionally, participants reported that they received increased emotional support from family members during lockdown, such as feeling important (44.9%), feeling be loved (44.8%), and caring for each other (47.5%).

Univariate regression analysis showed that age, place of residence, and mother employment status were significant predictors of changes in child maltreatment during lockdown. Older age was associated with a lower decrease in child maltreatment (b = −1.48, *p* < 0.001, 95% CI = −1.96, −1.01). Individuals who lived in a city had a lower decrease in child maltreatment than those who lived in the country (b = −1.29, *p* = 0.009, 95% CI = −2.26, −0.32). Mother employment was associated with a lower decrease in child maltreatment (b = −1.05, *p* = 0.037, 95% CI = −2.05, −0.06). Gender, household income, parental relationship, father employment, depression, anxiety, and psychotic symptoms were not significantly associated with decrease in child maltreatment.

### 3.2. Child Maltreatment Pre- and Post-Lockdown

Figure 1 presents the prevalence of each type of child maltreatment pre- and post-lockdown. Physical neglect (before: 24%, after: 26.3%) was the most common aspect of childhood trauma followed by emotional neglect (before: 20%, after: 20.6%), emotional abuse (before: 3.6%, after: 3.4%), physical abuse (before: 2.4%, after: 3.1%), and sexual abuse (before: 1.6%, after: 2.9%). Compared with before the lockdown, the prevalence of sexual abuse significantly increased 8 months after the lifting of lockdown, χ^2^(1) = 9.36, *p* = 0.002, OR = 1.82, 95% CI = 1.23, 2.70. No significant differences were found in physical abuse and neglect, emotional abuse and neglect, and witnessing domestic violence.

To investigate which risk factors were associated with changes within an individual’s sexual abuse, the linear mixed model was conducted controlling for age, emotional and physical neglect, emotional and physical abuse, and witnessing domestic violence. Mean scores rather than a binary outcome indicator were used for the outcome measure. As presented in Table 2, males showed a significantly larger increase in sexual abuse from pre- to post-lockdown than females. Depression, psychotic symptoms, and state anhedonia at baseline were significantly associated with an increase in sexual abuse. Simple effect analysis found that adolescents with high levels of depression, psychotic symptoms, and state anhedonia showed a larger increase in sexual abuse than those having low levels of these symptoms. Household income, parental employment and marital relationship, place of residence, social anhedonia, and anxiety were not significantly associated with an increase in sexual abuse.

Depression = the Beck Depression Inventory; Anxiety = The Generalized Anxiety Disorder 7-item scale; Psychotic experiences= The Brief 8 Item Community Assessment of Psychotic Experiences-Positive Scale; Social anhedonia = The Anticipatory and Consummatory Interpersonal Pleasure Scale; State anhedonia = the Snaith–Hamilton Pleasure Scale; Child maltreatment = The Childhood Trauma Questionnaire-Short Form. The model is controlled for age, emotional and physical neglect, emotional and physical abuse and witnessing domestic violence. Bonferroni *p*-value adjusted for multiple testing, *p* < 0.05.

## 4. Discussion

This longitudinal study investigated changes in child maltreatment pre-, during, and post-lockdown in Chinese children and adolescents and identified potential risk factors. Our results found an overall decrease in all types of child maltreatment including emotional and physical abuse, emotional and physical neglect, sexual abuse, and witnessing domestic violence during lockdown. Older age, individuals who lived in cities, and those with mother employed showed smaller decreases in child maltreatment during lockdown. However, 8 months after the easing of lockdown, the prevalence of sexual abuse (2.9%) significantly increased compared with pre-lockdown (1.6%). Physical abuse and neglect also showed an increasing but insignificant trend. These findings suggest that maltreatment in children and adolescents experienced short-term relief during lockdown and a long-term negative effect after lockdown. Therefore, the claims that lockdown restrictions would have a dramatic elevated risk for child abuse and neglect were unsupported by the current findings. In addition, being male, suffering from depression, state anhedonia, and having psychotic symptoms were associated with increased sexual abuse from pre- to post-lockdown.

Contrary to our expectations, more than half of children and adolescents reported no changes in maltreatment during lockdown in the present study. However, the percentages reporting a decrease were much larger than those reporting an increase, with decreases ranged from 18% to 47.5%, whereas increases ranged from 5.1% to 9.1%. This was consistent with prior research showing that there was a very mild increase, of 1–3.7%, in different types of child maltreatment during the pandemic in Hong Kong [10]. Although there were concerns that the containment measures would cause large increases in child maltreatment during lockdown [4,31], we did not find evidence supporting these predictions. Indeed, there was evidence showing violent crime figures, including sexual offenses and domestic violence in 2020, underwent little change [32] or even decreased, compared to forecasts [33]. It is plausible that family cohesion and closeness could be strengthened because family may become far more important following the global COVID-19 crisis [18]. For example, positive changes in parent–child relationships occurred during lockdowns, with an increase in everyday caregiving activities, positive parenting behavior, and better parent–child relationships [14,15]. People exposed to the SARS epidemic experienced positive parenting and high levels of parental support [17]. Another possibility is that normal factors that may influence mental health become less important in the aftermath of a natural disaster [34], as shown by the beneficial impacts of adversity on psychosocial functions, such as affiliative, cooperative, and trusting behaviors [35]. These favorable changes may have served as a cushion against the negative impacts due to economic stress and social isolation created by the COVID-19 pandemic. In addition, the slowing of daily routines may be beneficial to the wellbeing of parents and children because of reduced expectations and pressures, and the removal of the need to make difficult transitions between work/school and home, as vacations and weekends are periods of relaxation and stress reduction [36]. These potentially beneficial effects may also explain why positive psychological functioning, such as wellbeing, life satisfaction, or connectedness, were substantially unaffected by the COVID-19 lockdown among people in general [19]. Overall, despite the substantial effects of the pandemic on the lives of families with children, the capacity of the pandemic to enhance supportive parent–child relationships may have been a key factor in reducing child abuse and neglect during the lockdown period.

However, these observed positive changes did not last after lockdown. In the present study, 8 months after the lifting of lockdown, child sexual abuse showed an increased trend compared with before lockdown, suggesting that, although the immediate impact of lockdown on child maltreatment, may be positive, the long-term impact was negative. The increase may have been associated with a significant stressful home environment due to financial stress and unemployment when people returned to normality, post-lockdown. The impacts of these economic hardships on families may have been buffered by a slowing in the pace of life, lower expectations, and government financial support during lockdown. However, the reopening of society may have increased mental health issues, even leading to abuse toward families and children, if unemployment persisted for a long time after lockdown [37]. For example, prior to this pandemic, child abuse and neglect were greater in families with unemployed parents, with children experiencing four times the degree of neglect and twice the degree of physical abuse [38]. Maternal and paternal stress and hostile parenting increased the risk for physical abuse and neglect against children, with higher psychological aggression predicting an increase in abuse across time [39]. The destructive role of stress on parenting quality is noted throughout the child maltreatment literature and theories, such as the family stress model [40]. Children who were under stressful home environments may become distant, isolated, and insecure, and more susceptible to be targeted by a perpetrator at school or neighborhood [22]. A systematic review showed that sexual violence is one of the most prevalent types of violence post-disaster [1]. In addition, children’s own interpersonal and academic pressures about the transition from home to school may result in a range of negative mental health symptoms, including depression, anxiety, and psychotic symptoms [31], as these symptoms in our study had a significant increase that paralleled increased sexual abuse. In conjunction with the above observation, this therefore indicates that the shadow of lockdown increased the risk of a significant long-term negative impact on child maltreatment. There is an urgent need to scale up efforts to reduce sources of stress for caregivers and to support adolescent mental health with adaptations to post-pandemic arrangements to protect children from threats to their safety.

Consistent with previous studies [41,42], the current study found that multiple socio-economic and individual characteristic factors were significant risk factors of child maltreatment. During lockdown, individuals having an old age, living in cities, and having a mother employed experienced smaller decreases in maltreatment than those without these conditions. Previous studies reported younger children were significantly more at risk for child abuse [43]. However, in our present study, inconsistent findings were identified, as higher age was associated with greater maltreatment. It is possible that elder adolescents could benefit less from increased family time during lockdowns because they prefer to separate themselves from their parents to become autonomous individuals [44]. After the lifting of a lockdown, gender, adolescent depression, state anhedonia, and psychotic symptoms were important risk factors that significantly impacted the increase in sexual abuse from pre- to post-lockdown. It has been well established that those with pre-existing mental-health conditions are most vulnerable to family violence during the pandemic [1]. Somewhat surprisingly, no other pre-existing characteristics were found to significantly contribute to a difference in an individual’s sexual abuse after the lifting of lockdown; in particular, living in a low-income home or in an urban area, or having an unemployed father, would put a person at greater risk for child maltreatment. It is plausible that economic rapid recovery after lockdown in China may reduce home financial pressure. However, this remains an open question and an important challenge for future research.

The significant strength of our study is that it offers strong empirical data to assess the short-term and long-term effects of lockdown with respect to child maltreatment; nonetheless, there are some potential limitations. Although the sample was large, it was a convenient sample taken from local schools, which increased the chance of sampling bias. Moreover, retrospective child maltreatment assessments were used. However, given the acute nature of the pandemic, changes in child maltreatment were likely to be largely attributable to the events associated with lockdown restrictions. Using a retrospective and prospective design, we could take a snapshot of child abuse and neglect in a typical year and compare it to different timepoints throughout the lockdown. Secondly, some important risks, such as parental education, mental health status, family functioning, and neighborhood characteristics, were not collected in this study, which restricted us to explore the development of child sexual victimization. Given that many countries are still in their second or third COVID-19 lockdowns, and the responses of different countries to COVID-19 have varied widely compared to some of the strategies undertaken in China, both the short-term buffering and the long-term suffering effects reported here remain open to further investigation.

## 5. Conclusions

The present study systematically investigated changes in child maltreatment before, during, and after the COVID-19 lockdown. The findings showed that child maltreatment decreased during lockdown, but increased 8 months after the lifting of lockdown, indicating that the impact of lockdown on child maltreatment was beneficial in the short-term but detrimental in the long-term. In addition, potential risks including being male, suffering depression, state anhedonia, and experiencing psychotic symptoms were associated with increased sexual abuse. More research is needed to disentangle the mechanisms of the positive and negative effects of lockdowns on child maltreatment.

## Figures and Tables

**Figure 1 ijerph-19-03350-f001:**
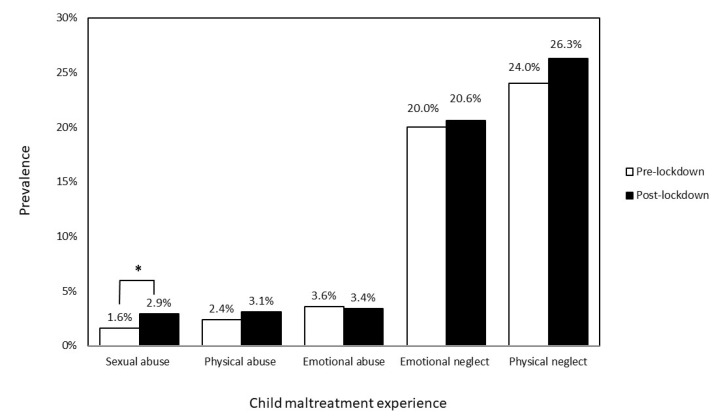
The prevalence of child maltreatment pre- and post-lockdown. * Bonferroni correction *p* < 0.05, data shows percentage and percentage error.

**Table 1 ijerph-19-03350-t001:** Self-rated changes (%) for the child maltreatment measure during lockdown.

Items	Increase (%)	Unchanged (%)	Decrease (%)
**Physical abuse**			
PA9-Hit hard enough to see a doctor	6.8	70.2	22.9
PA11-Hit hard enough to leave bruises	6.7	70.0	23.3
PA12-Punished with hard objects	6.5	69.5	24.0
PA15-Was physically abused	6.3	70.5	23.2
PA17-Hit badly enough to be noticed	7.0	71.4	21.6
PA29-Being burned by parents	5.1	72.8	22.1
PA30-Being threaten using knife by parents	5.2	72.7	22.2
PA31-Being choked by parents	5.3	72.5	22.2
**Emotional abuse**			
EA3-Called names by family	13.4	63.2	23.4
EA8-Parents wished was never born	9.1	67.5	23.4
EA14-Family said hurtful things	8.1	69.1	22.8
EA18-Felt hated by family	6.6	71.0	22.5
EA25-Was emotionally abused	6.1	71.9	22.0
**Sexual abuse**			
SA20-Was touched sexually	5.8	72.2	21.9
SA21-Hurt if didn’t do something sexual	5.5	72.5	22.0
SA23-Made to do sexual things	5.5	72.6	21.9
SA24-Was molested	5.7	72.4	21.8
SA27-Was sexually abused	5.3	72.7	22.0
**Emotional neglect**			
EN5-Made to feel important *	44.9	49.5	5.5
EN7-Family felted loved *	44.8	49.4	5.8
EN13-Was looked out for *	46.1	48.6	5.3
EN19-Family felt close *	37.3	56.2	6.6
EN28-Family was source of strength *	36.7	56.3	6.9
**Physical neglect**			
PN1-Not enough to eat	21.7	60.3	18.0
PN2-Got taken care of *	47.5	46.3	6.2
PN4-Parents were drunk or high	9.2	66.7	24.0
PN6-Wore dirty clothes	16.4	63.9	19.7
PN26-Got taken to doctor *	36.8	56.5	6.7
**Witnessing domestic violence**			
WDV32-parents language abuse	5.6	71.9	22.5
WDV33-parents physical abuse	5.4	72.6	22.1
WDV34-parents abused towards their sibling	5.5	72.2	22.4

* Reverse coded items, Child maltreatment = The Childhood Trauma Questionnaire-Short Form. To examine the effect of the COVID-19 lockdown on child maltreatment, the CTQ scale was modified to ask how changes in behaviours occurred during lockdown compared with before (1 = decreased, 2 = unchanged/same as before, 3 = increased).

**Table 2 ijerph-19-03350-t002:** Mixed model predicting an increase in sexual abuse from before to after lockdown.

	*B*	t	*p*	95% CI
Interactions with time (difference in change from pre-lockdown to 8 months after the lifting of lockdown)				
Place of residence (country)	0.02	1.03	0.31	−0.06, 0.06
Gender (female)	0.08	4.29	<0.001	0.04, 0.12
Parent’s marriage (married)	[Reference]	-	-	-
Separated	−0.01	−0.40	0.690	−0.08, 0.05
Divorce	−0.02	−0.23	0.816	−0.17, 0.13
House income (high)	[Reference]	-	-	-
Average	0.05	0.52	0.600	−0.15, 0.25
Low	0.04	1.39	0.165	−0.02, 0.10
Father employment (jobless)	−0.03	−1.20	0.231	−0.07, 0.02
Mother employment (jobless)	−0.02	−0.81	0.420	−0.06, 0.02
Depression	−0.08	−5.91	<0.001	−0.11, −0.05
Anxiety	0.01	0.78	0.434	−0.01, 0.03
Psychotic experiences	0.14	11.37	<0.001	0.12, 0.17
State anhedonia	0.03	3.18	0.001	0.01, 0.04
Social anhedonia	0.00	−0.26	0.795	−0.02, 0.02

## Data Availability

The data used to support the findings of this study are available from the corresponding author upon request.

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
