# Peer review of "The Short- and Long-Term Impact of COVID-19 Lockdown on Child Maltreatment"

_ijerph, 2022, doi:10.3390/ijerph19063350_

Round 1

Reviewer 1 Report

The subject of this paper is of clear significance in light of the impact of the pandemic and the modes of dealing with it have had on the lives of the population and particularly on children. It’s strength lies in its direct questioning of the children.  However, the methodology of the survey is very poorly elucidated and leaves the reader with many questions.  

There is no mention as to when the first request was made on the internet (after first lockdown? How long was the lockdown? At what time during the lockdown? What was the nature of the lockdown (confinement to house or some permission to leave homes. This varies greatly between countries).  Do we know where the children were from?  It says that the announcement about the survey was published through the schools—were the children in school at the time or It is also not clear when the second request was made.  There is no n reported for the number of replies, only a single reference for the number of children with a single reply.  There is no description of the population which answered the questionnaire. 

The authors state that “Depression, psychotic symptoms, and state anhedonia were significantly associated with change in sexual abuse”.  When were these symptoms measured:  during the lockdown questionnaire or at the post lockdown questionnaire? From the text, it is not clear.

The writing requires significant editing.  One example: lines 49-50 Job loss, social isolation and negative parenting behaviours were at high risk to engage in psychological and physical abuse [8-10].  Job loss etc do not engage in abuse.  Parents with…

Author Response

-There is no mention as to when the first request was made on the internet (after first lockdown? How long was the lockdown? At what time during the lockdown? What was the nature of the lockdown (confinement to house or some permission to leave homes? This varies greatly between countries).  Do we know where the children were from?  It says that the announcement about the survey was published through the schools—were the children in school at the time or it is also not clear when the second request was made.  There is no n reported for the number of replies, only a single reference for the number of children with a single reply.  There is no description of the population which answered the questionnaire. 

Response: Thank you for pointing to this important issue. It was placed under a 76-day Chinese national lockdown beginning January 23, 2020. During the strict lockdown, all schools and public transportation systems were closed. People are confined to their homes in which only one person per household was allowed to exit once each two days. The strict lockdown was lifted on April 8, 2020. Schools were resumed and children and adolescents backed to school. Our study started at the beginning of the academic term in September, when students had 5 months normal life after the lift of the national lockdown. After that, there was not another national lockdown in China except small lockdowns occasionally happened in some city according to dynamic zero-COVID policy.

The two-wave study was conducted among a convenient sample from three secondary schools at Changsha city in Hunan province. At baseline, students from all participating schools were invited to participate in the study on a voluntary basis. Of the 3273 invited students, 2821(86.19%) completed the questionnaire during the first survey from 22 September to 25 October 2020. Participants completed the child maltreatment history questionnaire relate to the lockdown and their mental health level at present. Eligible participants the second survey were students who had participated at baseline. 87.56% students (n=2470) responded to the survey from 29 December 2020 to 16 January 2021, when it was 8 months after lockdown. The following exclusion criteria were applied: (1) students and their guardians (n=452) were not willing to participate in the study; (2) 122 students were inclined to respond to the items in a similar pattern (e.g., chose the same answer within the whole questionnaire); (3) 94 participants answered twice and their second answers were excluded; (4) 141 participants were excluded because the completion time was shorter than 554 s (2.5th percentile) or longer than 4204 s (97.5th percentile).  There were 2464 (86%) valid participants in the final sample, which comprised of 1461 (59%) females and 1003 (41%) males aged from 12 to 18, with a mean age of 15.48 (SD =1.76).

Because the Chinese Government still recommended the public to minimize face-to-face interaction after lockdown, this study was completed through an online survey platform (www.wjx.com), with the relevant smartphone link having been pushed to students and their guardians via family-school communication app by the school mental health service. Clicking on the link to the survey guided potential respondents to a page that provided information about the purpose of the study, the nature of the questions, and two electronic informed consents for students and their guardians, respectively.

To reduce the potential possibility of parental intervention in reporting child maltreatment, the survey was conducted during a regularly scheduled class period for mental health activity at school. Before the start of the investigation, the details of the study were given again by psychological teacher to all responders. Participation was voluntary and no incentive reward was given. Anonymity was emphasized, but as the survey was longitudinal study, participants were asked to give the last four digits of their Student Identified number. It took participants about 30 minutes to complete the survey.

Child maltreatment history was retrospectively assessed pre, during and post-lockdown. At baseline, participants were asked to report their child maltreatment experience before lockdown via the Childhood Trauma Questionnaire. To assess changes in child maltreatment related to the lockdown, participants also were asked to report their maltreatment experience for each item in terms of unchanged, increased and decreased during lockdown as compared to before. At follow-up, participants were asked to report their child maltreatment experience the Childhood Trauma Questionnaire from the lifting of lockdown to present. Participants at baseline were also asked to retrospectively report depression, anxiety, and psychotic symptoms during lockdown.

-The authors state that “Depression, psychotic symptoms, and state anhedonia were significantly associated with change in sexual abuse”.  When were these symptoms measured:  during the lockdown questionnaire or at the post lockdown questionnaire? From the text, it is not clear.

Response: Thank you for your suggestion. We revised the sentence: Depression, psychotic symptoms, and state anhedonia at baseline were significantly associated with increase in sexual abuse.

-The writing requires significant editing.  One example: lines 49-50 Job loss, social isolation and negative parenting behaviours were at high risk to engage in psychological and physical abuse [8-10].  Job loss etc do not engage in abuse.  Parents with…

Response: Thank you for suggestion. We have corrected the sentence: Parents with job loss, social isolation and negative parenting behaviours were at high risk to engage in psychological and physical abuse. We have carefully corrected typing mistakes and grammatical errors throughout our manuscript.

Reviewer 2 Report

The authors have studied the impact of COVID-19 pandemic lockdown on child maltreatment in China. This is an interesting and timely study. My suggestions and comments follow.

  • “The COVID-19 pandemic, as a natural crisis” should be revised to read “as a public health crisis”
  • “It a 28-item retrospective” should be revised to read “It is a 28-item retrospective”
  • “Invitations were sent to students and their parents via family-school communication app by the school mental health service.” The authors need to provide more info about how the participants were recruited. How the study was presented to the participants? What was the response rate?
  • What was your inclusion/exclusion criteria?
  • “Cross-sectional univariate regression analysis was performed to determine which factors, including sociodemographic and retrospective mental health symptoms of the lockdown period, were associated with change in child maltreatment during lockdown.” Please provide more details of your regression model. What were your independent variables and covariates?
  • This manuscript would benefit from English editing by a subject-matter expert.

Author Response

The authors have studied the impact of COVID-19 pandemic lockdown on child maltreatment in China. This is an interesting and timely study. My suggestions and comments follow.

-“The COVID-19 pandemic, as a natural crisis” should be revised to read “as a public health crisis”

Response: Thank you for suggestion. We revised this sentence: The COVID-19 pandemic, as a public health crisis, has brought a new threat to child health and safety.”

-“It a 28-item retrospective” should be revised to read “It is a 28-item retrospective”

Response: Thank you. We corrected this error.

-“Invitations were sent to students and their parents via family-school communication app by the school mental health service.” The authors need to provide more info about how the participants were recruited. How was the study presented to the participants? What was the response rate?

Response: Thank you for suggestion. It was placed under a 76-day Chinese national lockdown beginning January 23, 2020. During the strict lockdown, all schools and public transportation systems were closed. People are confined to their homes in which only one person per household was allowed to exit once each two days. The strict lockdown was lifted on April 8, 2020. Schools were resumed and children and adolescents backed to school. Our study started at the beginning of the academic term in September, when students had 5 months normal life after the lift of the national lockdown. After that, there was not another national lockdown in China except small lockdowns occasionally happened in some city according to dynamic zero-COVID policy.

Because the Chinese Government still recommended the public to minimize face-to-face interaction after lockdown, this study was completed through an online survey platform (www.wjx.com), with the relevant smartphone link having been pushed to students and their guardians via family-school communication app by the school mental health service. Clicking on the link to the survey guided potential respondents to a page that provided information about the purpose of the study, the nature of the questions, and two electronic informed consents for students and their guardians, respectively. To reduce the potential possibility of parental intervention in reporting child maltreatment, the survey was conducted during a regularly scheduled class period for mental health activity at school. Before the start of the investigation, the details of the study were given again by psychological teacher to all responders. Participation was voluntary and no incentive reward was given. Anonymity was emphasized, but as the survey was longitudinal study, participants were asked to give the last four digits of their Student Identified number. It took participants about 30 minutes to complete the survey.

The two-wave study was conducted among a convenient sample from three secondary schools at Changsha city in Hunan province. At baseline, students from all participating schools were invited to participate in the study on a voluntary basis. Of the 3273 invited students, 2821(86.19%) completed the questionnaire during the first survey from 22 September to 25 October 2020. Participants completed the child maltreatment history questionnaire relate to the lockdown and their mental health level at present. Eligible participants the second survey were students who had participated at baseline. 87.56% students (n=2470) responded to the survey from 29 December 2020 to 16 January 2021, when it was 8 months after lockdown. The following exclusion criteria were applied: (1) students and their guardians (n=452) were not willing to participate in the study; (2) 122 students were inclined to respond to the items in a similar pattern (e.g., chose the same answer within the whole questionnaire); (3) 94 participants answered twice and their second answers were excluded; (4) 141 participants were excluded because the completion time was shorter than 554 s (2.5th percentile) or longer than 4204 s (97.5th percentile).  There were 2464 (86%) valid participants in the final sample, which comprised of 1461 (59%) females and 1003 (41%) males aged from 12 to 18, with a mean age of 15.48 (SD =1.76).

-Inclusion/exclusion criteria

Response: Thank you for your suggestion. Inclusion criteria: students from all participating schools were invited to participate in the study. The following exclusion criteria were applied: (1) students or their guardians (n=452) were not willing to participate in the study; (2) 122 participants were inclined to respond to the items in a similar pattern (e.g., chose the same answer within the whole questionnaire); (3) 94 participants answered twice and their second answers were excluded; (4) 141 participants were excluded because the completion time was shorter than 554 s (2.5th percentile) or longer than 4204 s (97.5th percentile). We add these into our revision.

-“Cross-sectional univariate regression analysis was performed to determine which factors, including sociodemographic and retrospective mental health symptoms of the lockdown period, were associated with change in child maltreatment during lockdown.” Please provide more details of your regression model. What were your independent variables and covariates?

Response: Thank you for your suggestion. Independent variables included sociodemographic and mental health symptoms (depression, anxiety and psychotic) during lockdown. Sociodemographic variables were gender, age, place of residence, household income, parental employment status, parental relationship. The total score of changes in child maltreatment related to lockdown was dependent viable.

At baseline, participants were asked to retrospectively report changes in child maltreatment experiences during lockdown, which the response scale was modified to ask how changes behaviours occurred “during lockdown” (1=decreased, 2=unchanged/same as before, 3=increased). The total score was changes in child maltreatment related to the lockdown. In addition, participants were also asked to retrospectively assess depression, anxiety, and psychotic symptoms during lockdown.

-This manuscript would benefit from English editing by a subject-matter expert.

 Response: Thank you. We have carefully corrected typing mistakes and grammatical errors throughout our manuscript.

Reviewer 3 Report

This paper investigates the short and long-term impact of COVID-19 lockdown on child maltreatment, using retrospective data from before, during, and after the lockdown.

The research topic is unarguably relevant and timely. The paper is beautifully written. The introduction provides adequate framing for the paper and a sufficient overview of the background to the research questions. The research problem is clearly articulated, with an appropriate rationale and justification of its importance. The literature review encompasses a comprehensive coverage of available appropriate and contemporary literature. It entails a critical analysis that further expounds on the research problem. The research design is clearly described, with an adequate justification for the choice of methods and a clear account of how evidence has been analyzed. It demonstrates that impeccable norms of good research practice have been upheld in the conduct of research and it allows reproducibility by other scientists. The empirical data presented in the article are adequate and the discussion is sufficiently detailed, indicating a depth of insight that provides a firm foundation for contributing knowledge.

Two minor notes:
In line 148, the word “used” is missing.
The title “4. Discussion” would be more appropriate if it was phrased “4. Discussion and conclusion”

Author Response

This paper investigates the short and long-term impact of COVID-19 lockdown on child maltreatment, using retrospective data from before, during, and after the lockdown.

The research topic is unarguably relevant and timely. The paper is beautifully written. The introduction provides adequate framing for the paper and a sufficient overview of the background to the research questions. The research problem is clearly articulated, with an appropriate rationale and justification of its importance. The literature review encompasses a comprehensive coverage of available appropriate and contemporary literature. It entails a critical analysis that further expounds on the research problem. The research design is clearly described, with an adequate justification for the choice of methods and a clear account of how evidence has been analyzed. It demonstrates that impeccable norms of good research practice have been upheld in the conduct of research and it allows reproducibility by other scientists. The empirical data presented in the article are adequate and the discussion is sufficiently detailed, indicating a depth of insight that provides a firm foundation for contributing knowledge.

Two minor notes:
In line 148, the word “used” is missing.
The title “4. Discussion” would be more appropriate if it was phrased “4. Discussion and conclusion”

Response: Thank you very much for your suggestion. We have carefully corrected typing mistakes and grammatical errors throughout our manuscript.

Round 2

Reviewer 2 Report

- "During Chinese lockdown, children and adolescents reported that emotional abuse and neglect, physical abuse and neglect, sexual abuse, and witnessing domestic violence significantly decreased , whereas sexual abuse significantly increased 8 months after the lifting of lockdown compared with pre-lockdown. Being male, depression, state anhedonia and psychotic symptoms were associated with increased sexual abuse after lockdown."

In the result section of the abstract, please add "percentage change" values along with "p values" for each one of the maltreatment types.

- "Being male, depression, state anhedonia and psychotic symptoms were associated with increased sexual abuse after lockdown." please specify that this is in comparison to before lockdown.

Author Response

Response: Thank you for your suggestion. We have revised abstract: During Chinese lockdown, children and adolescents reported that the proportions of decrease (range 18-47.5%) in emotional abuse and neglect, physical abuse and neglect, sexual abuse, and witnessing domestic violence were greater than that of increase (range 5.1-9.1%). Compared with before lockdown (1.6%), the prevalence of sexual abuse significantly increased 8 months (2.9%) after the lifting of lockdown (p=.002). Being male, depression, state anhedonia and psychotic symptoms at baseline were associated with increased sexual abuse after lockdown.